# Determination of Cannabinoids in Meat Products and Animal Feeds in Singapore Using Liquid Chromatography–Tandem Mass Spectrometry

**DOI:** 10.3390/foods13162581

**Published:** 2024-08-18

**Authors:** Jia En Valerie Sin, Ping Shen, Lifei Huang, Yuansheng Wu, Sheot Harn Chan

**Affiliations:** 1National Centre for Food Science, Singapore Food Agency, 7 International Business Park, Singapore 609919, Singapore; valerie_sin@sfa.gov.sg (J.E.V.S.);; 2Department of Food Science & Technology, National University of Singapore, 2 Science Drive, Singapore 117543, Singapore

**Keywords:** cannabinoid, hemp, liquid chromatography–tandem mass spectrometry (LC-MS/MS), feed, animal tissue

## Abstract

There has been a growing interest in the use of hemp as an animal feed ingredient considering its economic value and nutritional properties. However, there is a paucity of research regarding the safety of hemp-based animal feed currently. Thus, this raises safety concerns on the potential transfer of cannabinoids from hemp-based animal feed to animal products intended for human consumption and its health effects. As such, the detection and quantification of cannabinoids in meat and animal feeds would be desirable for monitoring purposes. In this study, a simple, rapid and sensitive method for the simultaneous quantification of four major cannabinoids (delta-9-tetrahydrocannabinol, cannabidiol, cannabinol and tetrahydrocannabinolic acid) in meat and animal feeds by liquid chromatography–tandem mass spectrometry (LC-MS/MS) was successfully developed and validated. The method was selective and sensitive, achieving limits of detection and quantification for the four cannabinoids from 5 to 7 µg/kg and 15 to 21 µg/kg, respectively. The overall recovery with matrix-matched calibration curves for the cannabinoids ranged from 87–115%. The coefficients of variation were between 2.17–13.38% for intraday precision and 3.67–12.14% for inter-day precision. The method was subsequently applied to monitor cannabinoids in 120 meat and 24 animal feed samples. No cannabinoid was detected, suggesting no imminent food safety concerns arising from the potential incorporation of hemp and by-products in animal feed and nutrition under the promotion of sustainable agricultural practices.

## 1. Introduction

Cannabinoids are a group of chemical compounds found in the cannabis plant. Cannabis (marijuana) and hemp both belong to the same plant species *Cannabis sativa* but are derived from different varieties. Cannabis contains high levels of delta-9-tetrahydrocannabinol (THC) (average 15–30% THC), which is the primary active phytoconstituent responsible for its behavioral and psychotropic effects. Conversely, hemp has low levels of THC (≤0.3% THC). The main cannabinoid present in hemp is cannabidiol (CBD) which is non-psychoactive [1,2]. Therefore, unlike cannabis, hemp generally does not have psychoactive or abuse potential. The hemp seeds and roots are also devoid of cannabinoids.

Globally, there has been an increasing interest in the use of hemp in animal feeds due to its purported benefits. Hemp has been reported to be a rich source of nutrients, where the seeds contain high levels of proteins, carbohydrates, polyunsaturated fatty acids, vitamins and minerals. Hemp seed cake or meal, a by-product obtained after the removal of high-value oils from seeds, also retains its high protein and nutritional profiles [3,4,5]. This makes it an economical and more attractive source of animal feed compared to conventional feed grains. At present, the European Food Safety Authority (EFSA) permits the use of hemp seeds as feed for all animal species. However, the regulatory landscape for hemp as a feed ingredient is vastly different in other countries. Hemp products are not approved as livestock feed ingredients in Canada and will require a separate application [6]. Similarly, the use of hemp or hemp products as animal feed is not permitted in Australia and New Zealand [7,8]. While the 2018 Agricultural Improvement Act (“Farm Bill”) legalized hemp production in the United States, the use of hemp in animal feed remains prohibited.

The reservations about the use of hemp as an animal feed ingredient stem from a paucity of studies demonstrating the absence of cannabinoid accumulation in animals fed with hemp-based feed [9]. Consequently, this raises concerns about the potential transfer of cannabinoid residues to animal products intended for human consumption. In an experiment performed on 80 cows fed with a 3250 mg daily dose of THC pellets for six consecutive days, the transfer rate of THC from feed to milk was approximately 0.15% [3]. AgriFutures Australia found that THC was detected at low levels in all measured tissues (liver, kidney fat, subcutaneous fat and meat) in sheep after feeding them with 28 or 56% of industrial hemp biomass for 42 days [10]. Similarly, Wenger Animal Nutrient and Technology Innovation Centre reported that THC and other cannabinoid residues in eggs, blood and tissues of laying hens fed with 10–30% hemp seed cake for 16 weeks were below the detection level of 0.0025% [11]. On the other hand, there have also been studies demonstrating the adverse effects of hemp-based feed on animal health. For example, a study conducted by the German Federal Institute for Risk Assessment (BfR) showed that feeding lactating dairy cows with cannabinoid-rich industrial hemp silage resulted in negative changes in animal health and behavior, such as bradycardia and unsteady gait. It also resulted in measurable levels of cannabinoids in the cow’s milk of up to 316 µg THC and 1174 µg CBD per kg milk. At this level, the acute reference dose for humans (1 µg THC per kg body weight) was exceeded in several consumer groups in exposure scenarios for milk and dairy product consumption [12].

Given the conflicting evidence in current tissue residue studies, it is crucial to investigate the occurrence of cannabinoids in meat products and animal feeds and to perform subsequent risk analysis if necessary. This is especially so for Singapore, a small city-state that imports more than 90% of food from diverse sources ranging from Europe to the United States to Asia [13]. While methods have been developed to quantify cannabinoids in beef tissues and bodily fluids as well as animal feeds, no studies have been conducted to date to evaluate the occurrence of cannabinoids in foods of various animal origins as well as animal feeds available commercially [9,14].

Therefore, the objective of this study was to develop and validate a liquid chromatography–tandem mass spectrometry (LC-MS/MS) method for the quantification of four cannabinoids (delta-9-tetrahydrocannabinol, cannabidiol, cannabinol and tetrahydrocannabinolic acid) in animal tissues. Following validation, the method was applied to screen for the presence of cannabinoids in imported meat products and animal feed used by Singapore’s local farms.

## 2. Materials and Methods

### 2.1. Chemicals and Reagents

HPLC grade acetonitrile was purchased from Fulltime Specialized Solvent and Reagent Co., Ltd. (Anqing, Anhui, China), while LC-MS grade methanol was from Elite Advanced Materials Sdn Bhd (Rawang, Selangor, Malaysia). ELGA PURELAB Option Q7 (ELGA LabWater, Woodridge, IL, USA) water purification system was used for ultra-pure water (resistivity > 18.2 MΩ·cm). Analytical grade formic acid and ammonium formate were purchased from Sinopharm Chemical Reagent Co., Ltd. (Shanghai, China). Analytical standards of the cannabinoids (delta-9-tetrahydrocannabinol, cannabidiol, cannabinol and tetrahydrocannabinolic acid) were gifts from Health Sciences Authority (HSA), Singapore. All standards were of ≥97% purity. The molecular formulas and structures of the compounds are shown in Table 1.

### 2.2. Preparation of Standard Solutions

A standard mix solution of delta-9-tetrahydrocannabinol (THC), cannabidiol (CBD), cannabinol (CBN) and tetrahydrocannabinolic acid (THCA) was prepared at 100 µg/mL in methanol. The solution was stored at −20 °C and stable for 1 year. This stock solution was further diluted to 1 µg/mL and 0.1 µg/mL using methanol and stable for 1 year and 6 months, respectively, at −20 °C.

### 2.3. Samples

Imported meat (pork, beef, lamb, chicken) and local animal feed samples were collected between February 2023 to November 2023. Muscles of the meat samples were homogenized using a blender and stored at −20 °C prior to analysis. Similarly, animal feed obtained from local farms was homogenized using a blender and stored at room temperature prior to analysis.

### 2.4. Sample Extraction

The sample extraction method was modified from a previous in-house method for screening hemp seed products. A total of 1 g of the homogenized tissue sample was weighed in a 50 mL polypropylene centrifuge tube, and 10 mL of 0.1% *v*/*v* formic acid in acetonitrile was added. For feed samples, pure acetonitrile was added instead. Following this, the sample was vortexed and shaken at about 420 rpm for at least 10 min using a linear shaker (Edmund Bühler GmbH, Bodelshausen, Germany). The tube was subsequently centrifuged (Thermo Fisher Scientific, Waltham, MA, USA) at 4000 rpm for 10 min at 10 °C. A total of 2 mL of the supernatant was then transferred into a 15 mL polypropylene centrifuge tube and evaporated to dryness under a continuous stream of nitrogen at 45 °C (Caliper Life Sciences, Hopkinton MA, USA). The dried extract was reconstituted with 500 µL of 2 mM ammonium formate and 0.2% *v*/*v* formic acid in water-acetonitrile mixture (10:90). The sample was then filtered using a 0.2 µm nylon membrane filter prior to LC-MS/MS analysis.

### 2.5. LC-MS/MS Analysis

LC-MS/MS analysis was conducted using SCIEX ExionLC™ AD coupled to Triple Quad 6500+ system (SCIEX, Framingham, MA, USA). The analytical method was adapted from a previous in-house screening method for hemp seed products. An Agilent Poroshell 120 EC-C18 column (100 mm × 2.1 mm, 2.7 µm) was used for chromatographic separation. The injection volume was 10 µL and the flow rate was 0.5 mL/min. Mobile phase A was 2 mM ammonium acetate and 0.2% *v*/*v* formic acid in water, while mobile phase B was 2 mM ammonium acetate and 0.2% *v*/*v* formic acid in water–acetonitrile mixture (10:90). The solvent gradient used was as follow: 15–95% B at 0–3 min, 95% B at 3–5 min. Re-equilibration time was 2.5 min, and the total run time was 8 min.

MS/MS acquisition was performed using both positive and negative electrospray ionization (ESI), and multiple reaction monitoring (MRM) was conducted for each target analyte. Operating conditions for ESI were as follows: ion source temperature at 500 °C, curtain gas pressure of 20 psi, IonSpray voltage (IS) of 4000 V (−4000 V for negative mode), ion source gas 1 (GS1) and 2 (GS2) pressures of 40 and 50 psi. Nitrogen was used as collision gas. At least two transitions were monitored for each analyte, with the most abundant transition used for quantification. THCA was monitored in negative ionization mode (ESI−), while the remaining cannabinoids were in positive ionization mode (ESI+). The MRM parameters (precursor ion, product ions, collision energy, dwell time, declustering potential, exit potential, cell exit potential, retention time) were optimized and set accordingly for each individual compound as summarized in Table 2. All acquisitions and data analyses were performed using Analyst Software Version 1.7.2 (AB Sciex Pte Ltd., Singapore).

### 2.6. Method Validation

The analytical method was validated in accordance with in-house quality procedures and International Conference on Harmonization (ICH) Guidelines [15]. The parameters evaluated were selectivity, linearity and range, accuracy, precision (intraday/repeatability and inter-day/intermediate precision), limit of detection (LOD) and limit of quantification (LOQ).

#### 2.6.1. Selectivity

Selectivity of the method was determined by comparing the analytical signals of the blank matrix samples (i.e., matrix without any target analytes) with the analytical signals of blank matrix samples spiked with known concentrations of the target analytes. The analyte peaks from the product ions in the extracted ion chromatograms of the spiked samples should also fully overlap.

#### 2.6.2. Linearity and Range

Matrix-matched standard calibration curves were prepared using blank matrix samples spiked with the target analytes at 6 concentration levels ranging from 2 to 50 µg/L. A plot of measurement response (*y*-axis) against concentration (*x*-axis) was constructed to determine linearity. Linearity was expressed as the coefficient of determination, r^2^, and a calibration curve with a goodness-of-fit for r^2^ ≥ 0.99 was accepted. The linear range was subsequently determined.

#### 2.6.3. Accuracy and Precision

Accuracy and precision were assessed at 3 concentration levels—low (20 µg/L), medium (35 µg/L) and high (50 µg/L) using blank samples pre-spiked with the target analytes at the respective concentrations. A minimum of 3 replicates was prepared at each concentration level. Accuracy was expressed in terms of recovery. Precision was evaluated at each of the 3 concentration levels within the same day (repeatability) and across 3 different days (intermediate precision) and expressed as coefficient of variation (CV%).

#### 2.6.4. Limit of Detection (LOD) and Limit of Quantification (LOQ)

Limits of detection (LOD) and quantification (LOQ) were established based on the standard deviation of the response and slope of the calibration curve.

The *LOD* and *LOQ* may be expressed as:$$LOD=\frac{3.3\sigma }{S}\;LOQ=\frac{10\sigma }{S}$$

*σ* is the standard deviation of the response, while *S* refers to the slope of the calibration curve

#### 2.6.5. Matrix Effects

Matrix effects were evaluated by comparing the mean analyte peak areas in matrix fortified with the target analytes post-extraction and in standard solutions at 35 µg/L. Matrix effects were expressed as a ratio of the mean peak area of the analyte in post-extraction spiked samples to the mean peak area of the same analyte in standard solution, multiplied by 100.

## 3. Results and Discussion

### 3.1. Method Development

For the previous in-house screening method on hemp seed products, acetonitrile was used for extraction. However, 0.1% *v*/*v* formic acid in acetonitrile was used in this study instead for meat samples. This is because formic acid causes protein denaturation, thereby disrupting protein-drug binding such that the total amount of the drug can be extracted for analysis [16]. Given that protein-drug binding is not expected to occur in animal feed, acetonitrile was used as the extraction solvent instead. As the amount of cannabinoid residues expected to transfer from the hemp-based feed to animal tissues (if any) was low, the supernatant was evaporated to dryness and reconstituted in a lower solvent volume to concentrate the cannabinoid residues. In this study, the sample clean-up efficiency of 0.2 µm nylon membrane filter versus Oasis PRiME HLB SPE cartridge (Waters Corp, Milford, MA, USA) was investigated. Oasis PRiME HLB SPE cartridge reduces the matrix effect by removing interferences such as fats and phospholipids in the meat samples. On the other hand, a 0.2 µm nylon membrane filter helps in removing any particulate matter that is larger than 0.2 µm. Our results (summarized in Appendix A) showed that there was no significant difference in the intensities of the target analytes for 5 µg/kg and 10 µg/kg spiked samples processed using 0.2 µm nylon membrane filter versus Oasis PRiME HLB SPE cartridge. Given that one of the objectives of this study was to develop a simple analytical method, a 0.2 µm nylon membrane filter was selected as the sample clean-up method instead.

High-Performance Liquid Chromatography (HPLC) coupled with UV and photodiode array detection presents a relatively simple and inexpensive method for cannabinoid analyses. However, LC-MS/MS was selected in this study due to its superior analytical performance. Firstly, a literature search has shown that cannabinoid residues may be present in meat at low concentrations (as low as µg/kg levels), making LC-MS/MS a more appropriate technique given its higher sensitivity and lower detection limits [10]. Cannabinoids also often have structural isomers, which may pose a challenge to separate and quantify using HPLC-UV alone. However, LC-MS/MS identifies an analyte based on both chromatographic retention time as well as unique mass fragmentation pattern. Therefore, it can simultaneously differentiate and identify the compounds of interest in a single analysis even in a complex matrix like a meat sample. A C18 column was employed in this study for reversed-phase chromatographic separation. This is because cannabinoids are relatively non-polar compounds that will retain well by the C18 column, thereby allowing effective separation based on their hydrophobic interactions with the stationary phase and other structural properties. To determine the ionization mode to use in this study, MS spectra of the cannabinoids were studied in both positive and negative ion modes. Based on the analytical conditions used in this study, neutral cannabinoids (i.e., CBD, CBN and THC) ionized more readily in positive mode, while THCA is an acidic cannabinoid that had higher sensitivity and cleaner mass spectrum in the negative mode. As such, THCA was monitored in negative ionization mode (ESI-), while the remaining cannabinoids were in positive ionization mode (ESI+). The molecular ions and product ions of the cannabinoids were obtained from a literature search, and at least two product ions were selected for each cannabinoid based on the highest intensities and absence of interferences. The chemical structures and MS/MS spectra of the product ions can be found in Appendix A. The signal of each molecular ion to product ion transition was further optimized manually by altering the collision energy (CE). CE resulting in the highest signal intensity was chosen for each transition.

### 3.2. Method Validation

Comparison of the chromatograms of the matrix blank with that of the spiked matrix samples showed the absence of interfering peaks from endogenous compounds at the expected retention times of the cannabinoids in the blank samples (Figure 1). This indicates that the method was considered selective. The cannabinoids were eluted in the sequence of CBD, CBN, THC and THCA. CBD, CBN and THC are neutral cannabinoids, while THCA is an acidic cannabinoid. CBD might have been eluted first, as it has an additional polar hydroxyl group compared to CBN and THC. It might thus interact less strongly with the reversed-phase column used, resulting in the shortest retention time. This elution sequence is also consistent with other works utilizing similar analytical conditions [14,17,18,19,20].

The linearity, range, accuracy, precision, LOD and LOQ validation results of the four cannabinoids are presented in Table 3. All four cannabinoids displayed excellent linearity in the concentration range of 2–50 μg/kg, with r^2^ ≥ 0.99. The results indicated satisfactory accuracies for all the cannabinoids with a range of 87–115%. Accuracies at 20, 35 and 50 µg/kg for THC, CBD, CBN and THCA were in the ranges of 87–109%, 98–115%, 94–115% and 96–108%, respectively. The coefficients of variation were between 2.17–13.38% for intraday precision and 3.67–12.14% for inter-day precision. Given that the results did not exceed ±15%, it signifies good repeatability and intermediate precision of the method. The LOD refers to the lowest analyte concentration that can be reliably differentiated from background noise, while the LOQ is the lowest analyte concentration that can be reliably quantified. LODs ranged from 5–7 µg/kg, while LOQs ranged from 15–21 µg/kg. Chromatograms of the matrix blank, spikes at LOD and LOQ levels are shown in Figure 2. Previous studies indicated the presence of THC in meat at levels as low as 14.3 µg/kg following the feeding of hemp-based pellets [10]. Given that the LODs in this method are lower than the lowest detected concentration reported in the literature, it is less likely to result in any false negative detection.

For matrix effects, a value greater than 100% indicates ionization enhancement, while a value less than 100% indicates ionization suppression. Matrix suppression was observed for THC, CBD and CBN where the matrix effects were 58.8 ± 17.5%, 60.59 ± 13.3% and 48.1 ± 11.1%, respectively. On the other hand, matrix enhancement was observed for THCA at 158.5 ± 17.0%. The matrix effects observed were expected given that the method was more generic, serving a high-throughput purpose, with minimal sample clean-up. As such, endogenous compounds present in the matrix could be non-selectively extracted alongside the analytes to suppress/enhance the ion intensity of the target analyte. Deuterated internal standards were not used in this study which could have further contributed to the variations caused by matrix effects.

Overall, the results indicated that the analytical method was suitable for the quantification of cannabinoids in this study. Given that the total run time for each sample was only 5 min, it also demonstrates the capability of the method for rapid analysis of cannabinoids.

### 3.3. Screening of Meat and Feed Samples for Cannabinoids

In this study, a total of 120 imported meat (pork, beef, lamb, chicken) and 24 animal feed samples were randomly collected between February 2023 to November 2023. The breakdown of the types and sources of samples screened is shown in Figure 3.

Beef, chicken, pork and lamb were chosen as the matrices of interest, as these are the most commonly consumed meat products in Singapore, and there are studies and reports on the integration of hemp seeds and by-products in livestock nutrition. Seafood was not included in this study due to the lack of literature reporting the use of hemp and by-products in aquaculture. The selection and scale of sampling by food type was commensurate with Singapore’s import categories, volume and sources, where chicken had the highest import quantities, followed by pork, beef and lamb, and the top sources by country of origin are Australia and Brazil and the United States for beef, Brazil for pork, Australia and New Zealand for lamb [13]. Targeted sampling was conducted for chickens from Thailand due to reports of farmers researching the use of marijuana in chickens to replace antibiotics [21]. This accounts for the significant number of chicken samples from Thailand despite it not being a major source of chicken for Singapore.

The samples were evaluated for the presence of cannabinoids using the validated LC-MS/MS method described above. Screening results showed that cannabinoid residues were not detected in any of the 120 meat samples, indicating no food safety risk. The results were not surprising, given that EFSA is the only regulatory authority that has permitted the use of hemp seeds as feed ingredients for all animal species to date. Therefore, the use of hemp-based animal feed in other parts of the world is unlikely to be prevalent. For example, the planted area for all utilizations of industrial hemp in the US in 2023 was 27,680 acres [22]. Despite the large-scale production of industrial hemp, none of the meat samples from the US tested positive for cannabinoids. This suggests the absence of the illegal use of hemp as feed in food-producing animals. Given that Europe is not a major import source for Singapore, it is unlikely for our imported meat samples to be derived from animals fed with hemp-based feed. Being an import-reliant city-state, Singapore has also established a robust regulatory framework to ensure that the imports are safe. In this framework, only accredited sources that meet Singapore’s food safety and animal health standards will be allowed to export their food and products to Singapore.

To conduct a more holistic assessment of the current food safety risk associated with the use of hemp-based feed, feed samples obtained from local farms were also investigated. Similar to the results obtained for the meat samples, none of the feed samples screened contained cannabinoids. As mentioned above, the use of hemp-based animal feed is prohibited in most countries, and hence, the results are not unexpected. Furthermore, animal feed for food-producing animals is also strictly regulated in Singapore, where a permit is required for importation and a license is required for local manufacturing and processing of animal feed and ingredients [23].

### 3.4. Limitations

To account for matrix effects, a matrix-matched blank and spiked samples were included during each run as negative and positive controls in this study. However, deuterated internal standards for each cannabinoid (e.g., CBD-d_3_, THC-d_3_, THCA-d_3_) could be added in the future to further minimize variations resulting from matrix effects. This is recommended, given that results from this study showed that matrix suppression was observed for CBD, CBN and THC, while matrix enhancement was present for THCA.

To further ascertain the reliability of this method under different conditions, its robustness could be investigated in the future. This involves deliberately varying the method parameters (mobile phase composition/pH, column temperature, etc.) and environmental factors (room temperature, humidity, etc.) and monitoring the effects on the results obtained.

While the current validated method seems appropriate for the quantification of cannabinoids in animal tissues and feed, not all food matrices were included (e.g., egg, milk). Due to the possible transfer of cannabinoids from hemp-based feed into these matrices, they should be considered for future work. The levels of THC reported in the literature for milk are also lower than the LOD of this method (1.4–5 µg/kg), thus suggesting the need for further method optimization [12].

In a similar vein, only poultry feed was investigated in this study. Singapore’s agri-food sector comprises hen shell egg, seafood and vegetable farms. However, fish feeds were not included in this study due to the lack of reports and convincing evidence on the effectiveness and wide use of hemp-based feed for aquaculture. This represents a potential monitoring gap that could be addressed in future studies since there has been increasing interest and research on the use of hemp as an alternative feed ingredient in aquaculture [24,25].

## 4. Conclusions

This study describes the development and validation of a rapid and sensitive LC-MS/MS method for the quantification of four cannabinoids (delta-9-tetrahydrocannabinol, cannabidiol, cannabinol and tetrahydrocannabinolic acid) in meat and animal feed. The method was subsequently applied to the screening of imported meat products and feed from poultry farms in Singapore for the presence of cannabinoids. Results showed that none of the tested products contained cannabinoids, implying no imminent food safety risk to consumers in Singapore. Based on available information, this is the first study to develop an LC-MS/MS method for the quantification of cannabinoids in animal tissue and feed matrices and to monitor the presence of cannabinoids in imported meat products and animal feed used by Singapore local farms. Despite the encouraging results obtained in this study, continuous monitoring and risk evaluation of cannabinoids in meat and animal feed should be conducted under the growing global interest in the integration of hemp and by-products in animal nutrition to promote sustainable agricultural practices.

## Figures and Tables

**Figure 1 foods-13-02581-f001:**
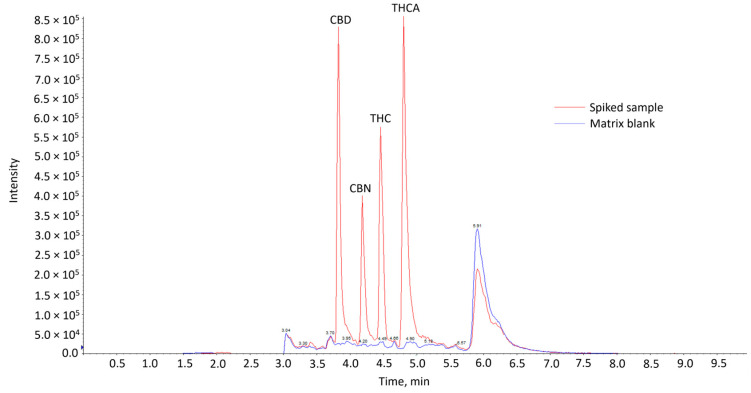
Overlay of LC-MS/MS chromatograms of matrix blank and spiked sample at 20 µg/kg.

**Figure 2 foods-13-02581-f002:**
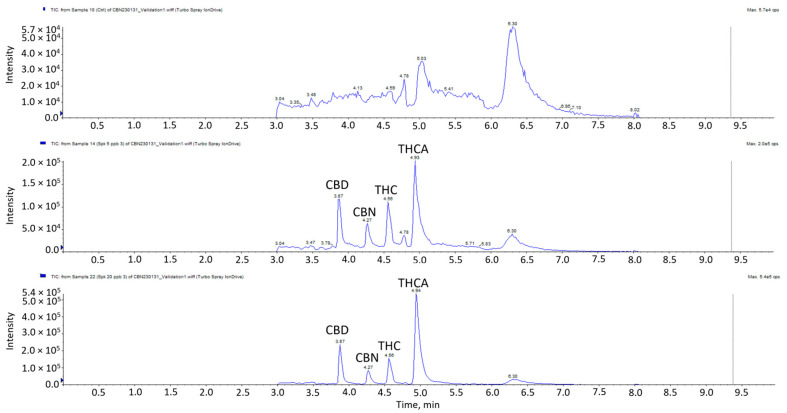
LC-MS/MS chromatogram of blank matrix spiked samples at LOD and LOQ.

**Figure 3 foods-13-02581-f003:**
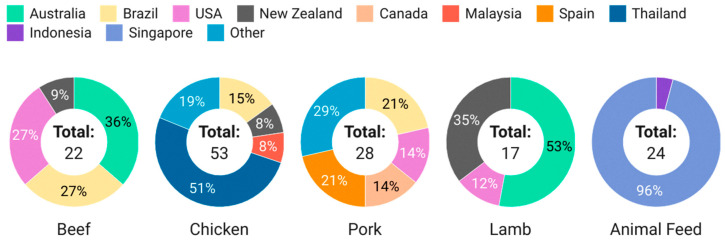
Breakdown of meat and feed samples screened in this study.

**Table 1 foods-13-02581-t001:** Chemical structures of 4 cannabinoids (delta-9-tetrahydrocannabinol, cannabidiol, cannabinol and tetrahydrocannabinolic acid).

Cannabinoid	Molecular Formula (Molecular Weight)	Chemical Structure
Delta-9-tetrahydrocannabinol (Δ^9^-THC)	C_21_H_30_O_2_ (314.46)	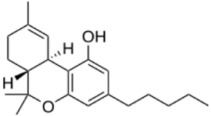
Cannabidiol	C_21_H_30_O_2_ (314.47)	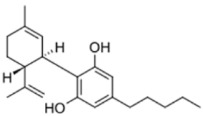
Cannabinol	C_21_H_26_O_2_ (310.43)	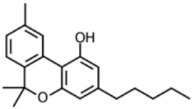
Tetrahydrocannabinolic acid	C_22_H_30_O_4_ (358.48)	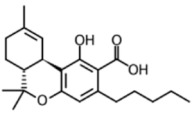

**Table 2 foods-13-02581-t002:** Multiple reaction monitoring (MRM) parameters and retention times of cannabinoids.

Analyte	Retention Time (min)	Precursor Ion (*m*/*z*)	Product Ion (*m*/*z*)	Dwell Time (ms)	Declustering Potential (V)	Exit Potential (V)	Collision Energy (eV)	Cell Exit Potential (V)
Delta-9-tetrahydrocannabinol (THC)	4.44	315.2	193.2	100	40	10	36	4
			259.2				27	
			123.1				35	
Cannabidiol (CBD)	3.81	315.1	193.2		40	10	29	4
			259.2				25	
			123.3				47	
Cannabinol (CBN)	4.17	311.4	223.3		40	10	29	4
			241.5				23	
Tetrahydrocannabinolic acid (THCA)	4.79	357.0	191.2	50	−100	−10	−42	−10
			245.0		−230		−42	
			179.0		−135		−40	

**Table 3 foods-13-02581-t003:** Validation results of 4 cannabinoids (delta-9-tetrahydrocannabinol, cannabidiol, cannabinol and tetrahydrocannabinolic acid).

Cannabinoid	Linear Range (µg/kg)	Linearity (R^2^)	LOD (µg/kg)	LOQ (µg/kg)	Spike Level (µg/kg)	Accuracy (%)	Precision	
							Intraday	Inter-day
							CV (%)	CV (%)
Delta-9-tetrahydrocannabinol (Δ9-THC)	2–50	0.990	6.86	20.77	20	109	7.54	7.57
					35	97	4.01	4.91
					50	87	13.38	12.14
Cannabidiol		0.990	6.98	21.14	20	115	11.86	7.52
					35	98	7.70	5.74
					50	98	6.65	5.00
Cannabinol		0.993	5.77	17.50	20	115	6.67	6.88
					35	102	4.62	6.59
					50	94	13.09	10.18
Tetrahydrocannabinolic acid		0.995	4.77	14.45	20	108	3.67	5.95
					35	103	2.17	5.05
					50	96	4.73	3.67

## Data Availability

The original contributions presented in the study are included in the article/Appendix A, further inquiries can be directed to the corresponding author.

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
