# Peer review of "Determination of Cannabinoids in Meat Products and Animal Feeds in Singapore Using Liquid Chromatography–Tandem Mass Spectrometry"

_foods, 2024, doi:10.3390/foods13162581_

Round 1
Reviewer 1 Report
Comments and Suggestions for Authors
The manuscript titled "Quantification of cannabinoids in meat products and animal feeds in Singapore using liquid chromatography-tandem mass spectrometry" presents a straightforward and efficient method for simultaneously measuring four key cannabinoids in meat and animal feeds. This is achieved through the use of liquid chromatography-tandem mass spectrometry (LC-MS/MS), which allows for rapid and sensitive analysis. The approach exhibited selectivity and sensitivity, resulting in detection and quantification limits for the four cannabinoids ranging from 5 to 7 µg/kg and 15 to 21 µg/kg, respectively. No traces of cannabinoids were found, indicating that there are no immediate food safety issues associated with the possible use of hemp and its by-products in animal feed and nutrition as part of sustainable farming methods. The devised technique may effectively identify the presence of hemp or its derivatives in animal feed. Nevertheless, my primary worry revolves around the imperative nature of this research in examining the potential hazards associated with the utilization of hemp in animal feed.
Specific issues of interest include:
1. Kindly present substantiated evidence of the possibility of utilizing hemp or its derivatives in different areas.
2. The technique development optimization lacks a discussion of how each parameter influences the detection result.
3. Kindly provide the chemical structure of the four desired analytes.
4. Please provide a rationale for using HPLC-MS over HPLC alone to analyze these components. Do they exist in tiny amounts in these matrices? Kindly present substantiating evidence.
5. The validation of this approach should be conducted in all four animal matrices (pork, beef, lamb, chicken), encompassing the determination of the limit of detection (LOD), limit of quantification (LOQ), accuracy, and precision outcomes. Please provide additional information or resources to enhance them.
6. In order to obtain feed and meat samples, it is advisable to collect them mostly from regions where the use of cannabinoids is permitted rather than from places where cannabinoids are prohibited.
7. Table 2 should provide the findings for the four different matrices, providing further information about each.
8. What is the process for developing procedures for sample extractions? Insufficient information is available. Kindly provide and discuss them.
9. It is apparent that the matrix effect is present, and it is advisable to explore the use of internal standards or other methods to rectify the offset.
10. please justify the necessity of detecting the presence of hemp in feed production when most nations prohibit its use. Furthermore, the outcome indicates the absence of any significant component observed.
Author Response
Please see the attached "replies to reviewer 1"

Reviewer 2 Report
Comments and Suggestions for Authors
Authors of the manuscript entitled “Determination of cannabinoids in meat products and animal feeds in Singapore using liquid chromatography-tandem mass spectrometry” developed and validated a LC-MS/MS method for quantifying four cannabinoids within animal tissues for their later screening within imported meat products and animal feed used by Singapore local farms. The manuscript is valuable in its field. Few comments and suggestions are to addressed as follows:
1. Authors are advised to annotate about the pilot trials of the mobile phase development till reaching to their finally adopted gradient mixture/phase percentages. This would be highly informative for researchers.
2. Authors are advised to annotate for the chemical structures of the depicted product ions for each analyte within a schematic representation. Additionally, authors are requested to provide the Daughter ion MS/MS spectra for each analyte.
3. Figure 1. Authors are asked to annotate for the extra peak at 5.89 min.
4. Authors are asked to rationalize the depicted elution sequence of their investigated analytes based on their respective chemical structure differences.
5. Authors are advised to provide the MRM transitions and respective mass spectrometer voltage (DP, EP, CE, CXP) of each analyte.
6. Authors can provide the robustness validation of the adopted technique to assess the ability of their analytical method to remain unaffected by small variations in the method parameters (mobile phase composition, column age, column temperature, etc.) and influential environmental factors (room temperature, air humidity, etc.).
7. Reference section should be modified in accordance with the requirements of MDPI style.
Author Response
please see the attached "replies to reviewer 2"

Reviewer 3 Report
Comments and Suggestions for Authors
The manuscript entitled " Determination of cannabinoids in meat products and animal feeds in Singapore using liquid chromatography-tandem mass spectrometry” developed and validated a simple, rapid, and sensitive method for the simultaneous quantification of four major cannabinoids in meat and animal feeds using LC-MS/MS given the growing interest in using hemp as an animal feed ingredient. The study aims to subsequently investigate other derivatives such as milk.
The analysis method is well described and seems easily reproducible.
Most of the references are recent (within the last 5 years). However, I would expand the bibliography by searching for recent works on LC-MS/MS, such as: Xin Xu and Lisa A. Murphy “Simple and fast quantification of cannabinoids in animal feeds by liquid chromatography–tandem mass spectrometry” etc.
The text is well-written, but here are a few suggestions for improvement:
Line 77-78: standardize the formatting of text
Line 93: standardize the formatting of text
Line 102: adjust format
Line 127: In the gradient section, the information on the run duration and the re-equilibration time is missing.
Line 280: standardize the formatting of text
Figura 1: In the figure, the overlay of the two chromatograms is shown, but the file name only indicates the spiked sample. To make a comparison, include the correct overlay figure or the two separate chromatograms. Moreover, the spiked sample is at 50 ppb, as seen in the file name, but it is indicated as 20 ppb.
Author Response
Please see the attached "replies to reviewer 3"

Round 2
Reviewer 1 Report
Comments and Suggestions for Authors
Even though I have some reservations about this subject, I read the author's response to the detailed comments and think this version should be released. Anyway, I think it's worth publishing.
Author Response
Reviewer's comments (2nd round): Even though I have some reservations about this subject, I read the author's response to the detailed comments and think this version should be released. Anyway, I think it's worth publishing.
Author's reply: thank you so much for reviewing our manuscript and guiding us through the revision with constructive comments. We are very happy to know that you are agreeable to accept our explanations and recommend accepting the revised manuscript for publication.